

# Two-dimensional topological paramagnets protected by $\mathbb{Z}_3$ symmetry: Properties of the boundary Hamiltonian

Hrant Topchyan[1], Vasilii Iugov[2,3], Mkhitar Mirumyan[1], Tigran Hakobyan[1,4],
Tigran A. Sedrakyan[5]⋆ and Ara G. Sedrakyan[1]

**1** Alikhanyan National Science Laboratory, Yerevan Physics Institute, Yerevan, Armenia
**2** Simons Center for Geometry and Physics, Stony Brook University,
Stony Brook, NY, 11794-3636, USA
**3** C. N. Yang Institute for Theoretical Physics, Stony Brook University,
Stony Brook, NY, 11794-3636, USA
**4** Institute of Physics, Yerevan State University, Yerevan, Armenia
**5** Department of Physics, University of Massachusetts, Amherst, Massachusetts 01003, USA

⋆ tsedrakyan@umass.edu

## Abstract

We systematically study gapless edge modes corresponding to $\mathbb{Z}_3$ symmetry-protected topological (SPT) phases of two-dimensional three-state Potts paramagnets on a triangular lattice. First, we derive microscopic lattice models for the gapless edge and, using the density-matrix renormalization group (DMRG) approach, investigate the finite-size scaling of the low-lying excitation spectrum and the entanglement entropy. Based on the obtained results, we identify the universality class of the critical edge, namely the corresponding conformal field theory and the central charge. Finally, we discuss the inherent symmetries of the edge models and the emergent winding number symmetry. As a result, one-dimensional chains with this symmetry form a model that supports gapless excitations due to its tricritical symmetry. Numerically, we show that low-energy states in the continuous limit of the edge model can be described by conformal field theory (CFT) with central charge $c = 1$, given by the coset $SU_k(3)/SU_k(2)$ CFT at level k=1.



# 1  Introduction

Symmetry-protected topological (SPT) phases [1–10] are a relatively new concept in condensed matter physics. Over the recent years, there has been notable research activity in that direction [11–41]. These phases fundamentally differ from the conventional phases defined by local order parameters while simultaneously possessing a topological nature. The notion of protection by symmetry means the impossibility of a smooth transition between phases without breaking the symmetry. This protection is what gives SPT phases their distinctive and robust topological properties. SPT phases usually emerge in non-degenerate gapped quantum systems with some symmetry $S$ at zero temperature and are beyond the classical theory of phase transitions. Two states belong to different phases if they cannot be connected adiabatically (without closing the gap) while preserving the symmetry $S$ on the whole path. The distinction between phases becomes apparent once considered on the system with a boundary. In this case, the "non-trivial" phases exhibit gapless excitations, and the ground state of the "trivial" phase can be given as a tensor product of local states. In $d$-dimensional space, SPT phases are classified by the non-trivial elements of the $H^{d+1}(S, U(1))$ cohomology group.

Commencing with the conventional understanding of phases described by Landau theory [42, 43], a cornerstone for the emergence of distinct phases and phase transitions is the symmetry group associated with the order parameter and its corresponding finite-size scaling universality classes. Spontaneously or explicitly broken symmetry is the key feature in this conventional notion of phases. The conventional states of the matter, as well as the SPT states, demonstrate short-range entanglement [37], unlike the topologically ordered states, which are long-range entangled [44–58]. Generally speaking, phases with short-range entangled states can be classified as either symmetry-broken (within the scope of conventional theory), SPT, or capable of simultaneously hosting symmetry-breaking and SPT orders. A distinctive feature of SPT states is their role in enabling the emergence of symmetry-protected gapless boundary excitations. Those excitations frequently belong to non-standard statistics and are important as key elements in the foundation of topological quantum computation.

The concept describing the SPT phases allows their classification by the cohomology classes of the corresponding discrete symmetry groups. It was introduced and developed by X.G. Wen and co-authors [1, 6–8, 13, 14]. As a particular case of the cohomology classification, the problem of the complete classification of SPT phases in one spatial dimension was solved in, e.g., Refs. [6, 7, 59, 60]. This formalism allows an intuitive understanding of the variety and properties of the SPT phases. However, it lacks an explicit presentation of specific models or a precise way of manipulation to construct models of SPT phases based on a desirable symmetry and a known topologically trivial mode. The situation changed with the work of Levin and Gu [12], where the authors show an SPT modification of the $\mathbb{Z}_2$ paramagnetic quantum Ising model with gapless $\mathbb{Z}_2$ symmetry-protected edge states.

Recently, the approach of [12] was extended to the $(\times \mathbb{Z}_3)^3$ symmetric Potts model, and the corresponding critical boundary model was derived [61]. For that case, the SPT phases are classified by the cohomology group $H^3((\times \mathbb{Z}_3)^3, U(1)) = (\times \mathbb{Z}_3)^7$. A study was performed on one of the boundary models belonging to a group of phases with seven generators. It was suggested that the low-energy effective conformal field theory is equivalent to the coset model $SU_k(3)/SU_k(2)$ at the level $k = 2$.

According to the conventional definition, the SPT models possess the following properties: first, the system must exhibit a global symmetry $S$ that remains unbroken in all phases. The so-called "trivial" phase typically features a gapped spectrum and the simplest form of the Hamiltonian. The ground state is usually expressed as a direct product of different subsystem states. The non-trivial phases of SPT models generally feature gapless edge modes. The fact of phases being symmetry protected is implemented as the non-existence of a symmetric series of transformations $U^\alpha$ continuous on $\alpha \in [0, 1]$, with $U^0 = 1$ and $U^1$ being the transformation connecting the two states in different phases. An alternative description of various phases is related to the 't Hooft anomaly [62]. It is the obstruction to gauging the system's symmetry $S$. In one dimension, cohomology classes describing the phases can be identified with the emergence of a projective representation of the symmetry group. This concept can be generalized for higher dimensions. The system can be gauged with the modified representation.

In this article, we study the SPT phases of the pure $\mathbb{Z}_3$ paramagnetic Potts model, in contrary to [61], where $(\times \mathbb{Z}_3)^3$ was considered. Out of all the phases given by the group $(\times \mathbb{Z}_3)^7$, only the phases related to its single $\mathbb{Z}_3$ subgroup were investigated, which correspond to the diagonal $\mathbb{Z}_3$ symmetry subgroup of the initial $(\times \mathbb{Z}_3)^3$ symmetry. It was also mentioned that the remaining 6 generators $\mathbb{Z}_3$ will result in the same edge theory as in a system protected by a $\mathbb{Z}_3$ symmetry, which is the object of study of the current work. In this case, SPT phases are classified by cohomology group $H^3(Z_3, U(1)) = Z_3$. The objective is to systematically construct a model that features SPT characteristics, starting with the $\mathbb{Z}_3$ Potts paramagnet as its trivial phase. We explicitly define the one-dimensional edge Hamiltonian and study its properties, including spectrum, symmetries, and low-energy modes. We also show that the model has a hidden $U(1)$ "winding number" anomalous symmetry. Through the further numerical study of the finite-size scaling behavior of excitation gap, entanglement entropy, and Kac-Moody currents on edge states, we argue that the effective low-energy theory of the edge model is the coset CFT $SU_k(3)/SU_k(2)$ at level $k = 1$. The present work also enables the study of gauged $\mathbb{Z}_3$ SPT models, disclosing anyonic properties of excitations in the topologically ordered phase.

## 2 Symmetry protected topological phase and the edge Hamiltonian

We start with the three-state paramagnetic Potts model defined on a triangular lattice. The Hamiltonian is given by

$$H_0 = -\sum_p (X_p + X_p^\dagger), \tag{1}$$

with the sum taken over all sites of the triangular lattice and $X_p$ being the $\mathbb{Z}_3$ generators on the sites [61]. We also introduce on-cite $\mathbb{Z}_3$ operators $Z_p$ which obey commutativity relations $X_p Z_p = \varepsilon Z_p X_p$ with $\varepsilon = e^{\frac{2\pi i}{3}}$. As $\mathbb{Z}_3$ generators, they have eigenvalues $1, \varepsilon, \varepsilon^*$. The matrices for $X_p$ and $Z_p$ are explicitly given as

$$X_p = \begin{pmatrix} 0 & 0 & 1 \\ 1 & 0 & 0 \\ 0 & 1 & 0 \end{pmatrix}, \qquad Z_p = \begin{pmatrix} \varepsilon & 0 & 0 \\ 0 & 1 & 0 \\ 0 & 0 & \varepsilon^* \end{pmatrix}. \tag{2}$$

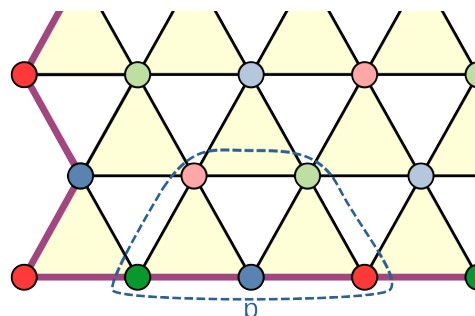

Figure 1: The lattice structure. Node coloring: red for $A$, green for $B$, and blue for $C$. The triangles are yellow-shaded for $\epsilon_\triangle = 1$ (pointing up) and not shaded otherwise ($\epsilon_\triangle = -1$, pointing down). A sample point $p$ is marked alongside the triangles $\triangle$, for which $\nu_{3\triangle}$ could affect operators at $p$. After the application, triangle terms $\nu_3$ reduce to link terms $\nu$ (Eq.12), and only the link terms marked purple remain in the final transformation (Eq.13).

$Z_p$ can also be represented as $Z_p = \varepsilon^{n_p}$, with $n_p$ having eigenvalues $0, \pm 1$. Commutativity relations take the form $X_p n_p = (n_p + 1) X_p$, where addition to $n_p$ should always be understood as by mod 3. The Hamiltonian Eq.1 has a global $\mathbb{Z}_3$ symmetry, given by operator

$$S = \prod_p X_p \,. \tag{3}$$

It is known that non-trivial SPT phases are defined by cohomology classes of corresponding symmetry group, that is $H^3(\mathbb{Z}_3, U(1)) = \mathbb{Z}_3$ in our case [6–8, 13]. The transformations of a topologically trivial Hamiltonian to Hamiltonians of non-trivial SPT phases are given by a unitary transformation generated by non-exact closed 3-forms (cocycles) of the symmetry group. The cocycle for $\mathbb{Z}_3$ symmetry was constructed in [61] and has the form $\psi_3(n_1, n_2, n_3) = n_1 n_2 n_3 (n_1 + n_2)$ in additive representation. The corresponding form used for the model construction is $\nu_3(n_a, n_b, n_c) = \varepsilon^{\psi_3(n_a, n_b - n_a, n_c - n_b)}$. The unitary transformation inducing the non-trivial edge modes is the product of $\nu_3$ terms over all of the lattice's triangles [18, 61, 63], with the lattice node states $n_p$ as its arguments as

$$U = \prod_{\triangle \equiv \langle abc \rangle} (\nu_{3\triangle})^{\epsilon_\triangle} = \prod_{\triangle \equiv \langle abc \rangle} \varepsilon^{\epsilon_\triangle n_a n_b (n_c - n_b)(n_b - n_a)} \,, \tag{4}$$

where $\triangle \equiv \langle abc \rangle$ denote every triangle in the lattice with $a$, $b$, and $c$ always being nodes of a specific corresponding color $A$, $B$, or $C$ (see Fig.1), and $\nu_{3\triangle}$ is the $\nu_3$ term corresponding to the given $\triangle$ triangle. In this formula, $\epsilon_\triangle = \pm 1$ indicates the orientation (pointing up or down) of the triangle $\triangle$.

The fact of $\psi_3$ being a closed form guarantees that $U$ is symmetric under $S$ in the bulk, however, the non-exactness ensures it is not symmetric on the boundary. We can write the $U$-transformed Hamiltonian as

$$H_a = U H_0 U^{-1} = -\sum_p (\bar{X}_p + \bar{X}_p^\dagger), \qquad \bar{X}_p = U_p X_p U_p^\dagger, \tag{5}$$

where $U_p$ is the part of $U$ that was generated by triangles containing the site $p$. In other words, it is the part of $U$ that is not commutative with $X_p$. We can define $\bar{n}_p = n_p$ and this way, the barred operators, $\bar{X}_p$, satisfy the same algebra relations as the initial ones. The Hamiltonian operator mentioned above can be split into bulk and boundary parts, which are still commutative. The symmetry of the boundary part, $H_{\partial,a}$, with respect to $S$, was broken by the proposed unitary transformation, but it can be restored straightforwardly by substituting

$$H_{\partial,a} \to H_\partial = \frac{1}{3}(H_{\partial,a} + S H_{\partial,a} S^\dagger + S^\dagger H_{\partial,a} S), \tag{6}$$

and the bulk part $H_{\partial*} = H_{\partial*,a}$ stays the same. Notice that the commutativity $[H_\partial, H_{\partial*}] = 0$ holds. By denoting $V = SUS^\dagger U^\dagger$ and $V' = S^\dagger USU^\dagger = SVS^\dagger V$, we can rewrite the edge Hamiltonian Eq.6 as

$$H_\partial = -\frac{1}{3}\sum_{p\in\partial}(\bar{X}_p + V\bar{X}_p V^\dagger + V'\bar{X}_p V'^\dagger) + \text{H.c.} \tag{7}$$

By using the relations $X_p \varepsilon^{kn_p} X_p^\dagger = \varepsilon^{k(n_p+1)}$ for $\forall k \in \mathbb{Z}$ one can obtain the expressions for $V$ and $V'$. Particularly,

$$V = \prod_{\langle abc\rangle} \varepsilon^{\epsilon_\triangle(-n_b^3 - n_b^2 + n_a n_b + n_b n_c - n_a n_c + n_a^2 n_b + n_b^2 n_c - n_a^2 n_c)}. \tag{8}$$

Now, we will focus on bringing $V$ to a simpler form, and then calculate the expression for $V'$ accordingly.

We note that $U_\triangle$ is not uniquely defined, and it can be changed without changing the SPT phase, thus allowing us to tweak $V$. This can be done by adding any 2-vertex terms (leading to a trivially identical transformation) or adding terms created by exact 3-forms to $U_\triangle$. The different 2-vertex terms are $f_{\alpha\beta}(n_p, n_q) = \varepsilon^{n_p^\alpha n_q^\beta}$ with $\alpha, \beta \in \{0, 1, 2\}$. It is straightforward to see that adding any of those terms to $U_\triangle$ creates an additional term $g_{\alpha\beta}^{pq} = Sf_{\alpha\beta}^{pq}S^\dagger f_{\alpha\beta}^{pq\dagger}$ to $V$ (we indicate the arguments of the functions as upper indices for more compact writing). The situation is simple because all $g_{\alpha\beta}^{pq}$ are commutative with initial $U_\triangle$ and with each other. The expressions of $g_{\alpha\beta}^{pq}$ are

| | $\alpha = 0$ | $\alpha = 1$ | $\alpha = 2$ |
|---|---|---|---|
| $\beta = 0$ | $1$ | $\varepsilon$ | $\varepsilon^{1-n_p}$ |
| $\beta = 1$ | $\varepsilon$ | $\varepsilon^{1+n_p+n_q}$ | $\varepsilon^{1+n_p-n_q-n_p n_q+n_q^2}$ |
| $\beta = 2$ | $\varepsilon^{1-n_q}$ | $\varepsilon^{1+n_q-n_p-n_p n_q+n_p^2}$ | $\varepsilon^{1-n_p-n_q+n_p n_q-n_p n_q^2-n_p^2 n_q+n_p^2+n_q^2}$ |

Adding different combinations of $f_{\alpha\beta}^{pq}$ terms to $U_\triangle$ can lead to the creation of basic elements in the exponent of $V$. Namely

$$\begin{aligned}
f_{(1)}^{pq} &= f_{10}^{pq} \to 1, \\
f_{(2)}^{pq} &= f_{(1)}^{pq}/f_{20}^{pq} \to n_p, \\
f_{(3)}^{pq} &= f_{(1)}^{pq}f_{(2)}^{pq}/f_{(2)}^{qp}f_{21}^{pq} \to n_p n_q - n_q^2, \\
f_{(4)}^{pq} &= f_{(1)}^{pq}/f_{(2)}^{pq}f_{(2)}^{qp}f_{(3)}^{pq}f_{(3)}^{qp}f_{22}^{pq} \to n_p^2 n_q + n_p n_q^2.
\end{aligned} \tag{9}$$

Using these simpler expressions, it is easy to see that multiplying $f_{(2)}^{bc}f_{(3)}^{ab}f_{(3)}^{cb}f_{(4)}^{ac}$ to $U_\triangle$ will bring $V$ to a simpler, color-rotation-invariant form:

$$V \to \prod_{\langle abc\rangle} \varepsilon^{\epsilon_\triangle(n_a n_b(n_a-1)+n_b n_c(n_b-1)+n_c n_a(n_c-1))} = \prod_{\triangle,\gamma} \varepsilon^{\epsilon_\triangle n_\gamma^\triangle n_{\gamma+1}^\triangle(n_\gamma^\triangle-1)}, \tag{10}$$

where $n_\gamma^\triangle$ is the spin of the vertex with color $\gamma$ in triangle $\triangle$. $\gamma$ runs through the colors and adding 1 to $\gamma$ changes it as the color changes when passing through points $a \to b \to c \to a$. Further addition of $f_{(3)}^{ca}f_{(3)}^{cb}/f_{(3)}^{ba}f_{(3)}^{ab}$ eliminates the (-1) in the braces and another $f_{(4)}^{ab}f_{(4)}^{bc}f_{(4)}^{ca}$ leaves us with the final form

$$V = \prod_{\triangle,\gamma} \varepsilon^{-\epsilon_\triangle n_\gamma^\triangle n_{\gamma+1}^\triangle(n_\gamma^\triangle-n_{\gamma+1}^\triangle)}, \tag{11}$$

which will be used in Eq.7 for the edge Hamiltonian. This expression has a mixed symmetry with regards to color permutations (it is symmetric under rotation and antisymmetric under reflection). We will later see the usefulness of such a property when writing the edge Hamiltonian.

The exact 3-forms aren't particularly helpful for our case, nevertheless it is useful to explicitly see their contribution to the expression for $V$ (Eq.11). The generators of exact 3-forms are written as $\delta\psi_2^{\alpha\beta}(n_1, n_2, n_3)$ where $\psi_2^{\alpha\beta}(n_1, n_2) = n_1^\alpha n_2^\beta$ and $\delta$ is the coboundary operator, which is explicitly known. The 3-forms themselves that should be written in $U_\triangle$ are $f'_{\alpha\beta}(n_a, n_b, n_c) = \varepsilon^{\delta\psi_2^{\alpha\beta}(n_a, n_b - n_a, n_c - n_b)}$. The multiplier added to $V$ as a result is $g'_{\alpha\beta} = Sf'_{\alpha\beta}S^\dagger f'^\dagger_{\alpha\beta}$. Here again, we use the commutativity relations of $g'$ operators. The results for $g'$ are given in the following table:

| | $\alpha = 0$ | $\alpha = 1$ | $\alpha = 2$ |
|---|---|---|---|
| $\beta = 0$ | 1 | $\varepsilon^*$ | $\varepsilon^{n_b - 1}$ |
| $\beta = 1$ | 1 | 1 | $\varepsilon^{n_a n_b + n_b n_c - n_a n_c - n_b^2}$ |
| $\beta = 2$ | 1 | $\varepsilon^{-n_a n_b - n_b n_c + n_a n_c + n_b^2}$ | $\varepsilon^{n_a^2(n_b - n_c) + n_b^2(n_a + n_b + n_c) - n_a n_b - n_b n_c + n_a n_c + n_b^2}$ |

This result doesn't provide additional freedom of transformation selection. However, some otherwise complicated transformations are in plain view here. Notice that we do not have the freedom to choose $p$ and $q$ here. After fixing the transformation $V$, the explicit expression of the aforementioned $V' = SVSV^\dagger$ should be calculated accordingly.

What remains is to find the result of the action of $V$ and $V'$ on $\bar{X}_p$ of the edge. All terms in $V$-s depend on the spins of two neighboring sites. Although there were 3-spin dependent terms in $U$, there was no way they could have made their way up here, as it would mean that those factors would not cancel out in bulk. There were also single-spin-dependent terms in our intermediate calculation that could not have vanished. We will discuss how to handle terms like that in a moment. Once again, we should note that only the parts of $V$ and $V'$ generated by the triangles that contain $p$ are important. It is also easy to see that 2-spin terms that depend on a vertex in bulk will cancel out, as they exist in two neighboring triangles with opposite signs. So $V$-s reduces to border link terms. Let us denote the neighboring edge vertices of $p$ by $p \pm 1$ with a specific positive traversal orientation (clockwise). We choose $\epsilon_\triangle = 1$ for the triangles which have ascending $\gamma$ along the traversal with that orientation, and $\epsilon_\triangle = -1$ for the others. So $\gamma_{p+1} = \gamma_p \pm 1$ implies $\epsilon_{\triangle_{p,p+1}} = \pm 1$. Using denotation $v(\alpha, \beta) = \varepsilon^{\alpha\beta(\beta - \alpha)}$, $V$ can be rewritten as

$$V_\partial = \prod_{p \in \partial} v(n_p, n_{p+1}). \tag{12}$$

The 1-spin terms can be handled in the final transformation. One can see that terms with $n_p$ cancel out if there is an even number of triangles containing $p$, which is always the case for $p$-s in bulk. In case of an odd number of containing the triangles, the sign of the term will depend on $\epsilon_{\triangle_{p,p-1}} = \epsilon_{\triangle_{p,p+1}}$. So, for our $v$ to handle such terms, $v$ should be split into two $v_+$ and $v_-$, depending on the orientation of the triangle. To satisfy the conditions mentioned above, the 1-spin term in $v_+$ should be the conjugate of one in $v_-$. Notice that the term $n_p$ will have come only from one triangle, as the rest will cancel out each other, so if we intend to include it in our new $v_\pm$, we should do it with an opposite sign than in $V$, as there will be two terms of that kind for the same $p$. The factorization for $V'$ corresponding to Eq.12 will be

$$V'_\partial = \prod_{p \in \partial} v'(n_p, n_{p+1}) = \prod_{p \in \partial} v^\dagger(n_p, n_{p+1}), \tag{13}$$

with $v'(n_p, n_{p+1}) = \varepsilon^{-(\alpha\beta - \alpha - \beta - 1)(\beta - \alpha)}$. The products over $v'$ and $v^\dagger$ are equal, as $v'(\alpha, \beta)$ and $v^\dagger(\alpha, \beta)$ differ only by terms $\propto \alpha - \beta$ and $\propto \alpha^2 - \beta^2$, and those contributions from neighbor

links cancel each other out. These lead to the following expressions for the action on $\bar{X}_p$-s:

$$
\begin{aligned}
V\bar{X}_p V^\dagger &= \varepsilon^{n_{p-1}(n_{p-1}+n_p-1)}\bar{X}_p \varepsilon^{-n_{p+1}(n_{p+1}+n_p+1)}, \\
V'\bar{X}_p V'^\dagger &= \varepsilon^{-n_{p-1}(n_{p-1}+n_p-1)}\bar{X}_p \varepsilon^{n_{p+1}(n_{p+1}+n_p+1)}.
\end{aligned}
\tag{14}
$$

As a result, we get a Hamiltonian $H_\partial$ that is translation invariant and does not depend on the shape of the edge:

$$
H_\partial = -\frac{1}{3}\sum_{p\in\partial}\bar{X}_p\left(\frac{1}{2}+\varepsilon^{(n_{p-1}-n_{p+1})(n_{p-1}+n_p+n_{p+1}+1)}+\text{H.c.}\right)+\text{H.c.}
\tag{15}
$$

An alternative way to write $H_\partial$ is

$$
H_\partial = -\sum_{p\in\partial}\bar{X}_p\,\delta_{(n_{p-1}-n_{p+1})(n_{p-1}+n_p+n_{p+1}+1)}+\text{H.c.}
\tag{16}
$$

Here $\delta_n$ is a function that is equal to 1 if $n$ is divisible by 3 and is equal to 0 otherwise. This becomes apparent after the observation that $\delta_n$ can be written as $3\delta_n = 1 + \varepsilon^n + \varepsilon^{-n}$.

The obtained boundary model was produced by the unitary transformation $U$ given by Eq.4. The other non-trivial boundary model results from $U^2 = U^\dagger$. However, the corresponding induced model will be given by a Hamiltonian of the same form as in Eq.15. This comes from the fact that the edge transformation operators (Eqs.12-13) satisfy $V'_\partial = V_\partial^\dagger$, thus considering $U^\dagger$ instead of $U$ will simply interchange $V_\partial$ and $V'_\partial$ while leaving the overall Hamiltonian the same. Together with the trivial boundary, those two non-trivial boundaries form the full cohomology group $H^3(\mathbb{Z}_3, U(1)) = \mathbb{Z}_3$, implying that this single Hamiltonian describes all the non-trivial edge theories.

The overall formulation procedure described above is more straightforward compared to [12, 61], and allows a further generalization to arbitrary symmetry groups.

## 3  Symmetries of edge Hamiltonian

Besides the translation and reflection symmetries, our Hamiltonian might have some additional symmetries. To discuss them, let us parametrize the Hamiltonian as

$$
\begin{aligned}
H_\partial(\lambda_1,\lambda_2,\lambda_3) = -\frac{1}{3}\sum_{p\in\partial}\bar{X}_p\Big[\lambda_1 &+ \lambda_2\varepsilon^{(n_{p-1}-n_{p+1})(n_{p-1}+n_p+n_{p+1}+1)} \\
&+ \lambda_3\varepsilon^{-(n_{p-1}-n_{p+1})(n_{p-1}+n_p+n_{p+1}+1)}\Big]+\text{H.c.},
\end{aligned}
\tag{17}
$$

with $H_\partial(1,1,1)$ being the boundary Hamiltonian we are dealing with.

From Eq.6 one can derive that the three parameters $\lambda_i$, $i=1,2,3$ will permute cyclically under action of operator $S$:

$$
SH_\partial(\lambda_1,\lambda_2,\lambda_3)S^\dagger = H_\partial(\lambda_3,\lambda_1,\lambda_2).
\tag{18}
$$

There is yet another global symmetry. Let us define the operator $\chi_p$ to be

$$
\chi_p = \begin{pmatrix} 0 & 0 & 1 \\ 0 & 1 & 0 \\ 1 & 0 & 0 \end{pmatrix} \text{ in the basis, where } \bar{X}_p = \begin{pmatrix} 0 & 0 & 1 \\ 1 & 0 & 0 \\ 0 & 1 & 0 \end{pmatrix} \text{ and } n_p = \begin{pmatrix} 1 & 0 & 0 \\ 0 & 0 & 0 \\ 0 & 0 & -1 \end{pmatrix}.
$$

It can be seen that $\chi_p \bar{X}_p^{\pm} \chi_p^{\dagger} = \bar{X}_p^{\mp}$ and $\chi_p n_p \chi_p^{\dagger} = -n_p$. We can define the product of all $\chi_p$-s in the system as an operator, $P$. Using the commutation relation between $\bar{X}_p$ and $n_p$, one can show that

$$PH_{\partial}(\lambda_1, \lambda_2, \lambda_3)P^{\dagger} = H_{\partial}(\lambda_1, \lambda_3, \lambda_2). \tag{19}$$

Now it is clear that the Hamiltonian has a global $\mathbb{S}_3$ symmetry with generators $S$ and $P$.

This extended space of a parametrized Hamiltonian is beyond the SPT configuration space, as in the case of unequal $\lambda$-s, the $\mathbb{Z}_3$ symmetry (Eq.3) is lost. In this new space, the permutation $S$ is a duality map between the strong and weak coupling regimes. The self-duality point is then the point of transition (critical point) between those regimes. In our case of three parameters, the self-"duality" point $\lambda_1 = \lambda_2 = \lambda_3$, which is the parameter value to get our boundary Hamiltonian, becomes a tricritical point. As for (at least) a second-order phase transition point, we now expect gapless excitations in our boundary model $H_{\partial}(1,1,1)$. It is important to note that the discussed criticality and transitions do not occur between the SPT phases, and are rather defined on an expended space, where our model corresponds to the critical point.

## 3.1 Winding number

Here we identify an additional "winding" symmetry of the edge Hamiltonian. To this end, we write it using a different parametrization of the basis states. Namely, we will parametrize them by the last spin $n_N$ and a set of differences $w_i$ defined by

$$\varepsilon^{w_i} = \varepsilon^{n_i - n_{i-1}}, \qquad w_i \in \{-1, 0, 1\}, \quad i = 1, \dots, N, \quad n_0 \equiv n_N. \tag{20}$$

Alternatively,

$$w_i = \delta_{n_i - n_{i-1} - 1} - \delta_{n_i - n_{i-1} + 1}, \qquad i = 1, \dots, N, \quad n_0 \equiv n_N. \tag{21}$$

This set of variables is not completely independent because $\varepsilon^{w_1 + \dots + w_N} = 1$. The Hamiltonian can be written in terms of $w_i$ as (see Eq.16)

$$H_{\partial} = -\sum_{p \in \partial} \bar{X}_p \delta_{(w_p + w_{p+1})(w_{p+1} - w_p + 1)} + \text{H.c.}, \tag{22}$$

$$H_{\partial} = -\sum_{p \in \partial} \bar{X}_p (\delta_{w_p - 1} \delta_{w_{p+1} + 1} + (1 - \delta_{w_p - 1})(1 - \delta_{w_{p+1} + 1})) + \text{H.c.} \tag{23}$$

Now, we introduce a new operator, $\mathcal{W}$, which, in terms of $w_p \in \{-1, 0, 1\}$ can be expressed as

$$\mathcal{W} = \frac{1}{3} \sum_{p \in \partial} w_p = \sum_{p \in \partial} (\delta_{w_p - 1} - \delta_{w_p + 1}). \tag{24}$$

The operator $\mathcal{W}$ counts the full number of turns that $\varepsilon^{n_p}$ makes around the unit circle as we move around the boundary $\partial$, so we will call $\mathcal{W}$ the "winding number" operator. When $n_{p+1} = n_p + 1$, we count that as a rotation by $\frac{2\pi}{3}$ and when $n_{p+1} = n_p - 1$, we count that as a rotation by $-\frac{2\pi}{3}$. One can check that the winding number operator $\mathcal{W}$ commutes with the Hamiltonian. The nontrivial part of that commutator calculation reduces to the observation that

$$\bar{X}_p (\delta_{w_p - 1} - \delta_{w_p + 1} + \delta_{w_{p+1} - 1} - \delta_{w_{p+1} + 1})(\delta_{w_p - 1} \delta_{w_{p+1} + 1} + (1 - \delta_{w_p - 1})(1 - \delta_{w_{p+1} + 1}))$$
$$= \bar{X}_p (\delta_{w_p + 1} - \delta_{w_p} + \delta_{w_{p+1}} - \delta_{w_{p+1} - 1})(\delta_{w_p - 1} \delta_{w_{p+1} + 1} + (1 - \delta_{w_p - 1})(1 - \delta_{w_{p+1} + 1}))$$
$$= (\delta_{w_p - 1} - \delta_{w_p + 1} + \delta_{w_{p+1} - 1} - \delta_{w_{p+1} + 1})\bar{X}_p (\delta_{w_p - 1} \delta_{m_{p+1} + 1} + (1 - \delta_{w_p - 1})(1 - \delta_{w_{p+1} + 1})), \tag{25}$$

which one can confirm by substituting all 9 possible combinations of $w_p$ and $w_{p+1}$. Thus, the winding number operator $\mathcal{W}$ is conserved by the Hamiltonian. The winding number symmetry

generated by $\mathcal{W}$ is a topological symmetry of the boundary model since it takes values in $\mathbb{Z}$ instead of $\mathbb{Z}_3$. It is a distinctive feature of the non-trivial boundary model, as long as an arbitrary stripe of nodes within the bulk doesn't have such a motion integral.

The symmetry can be reformulated in terms of a conserved current $j_p^\mu = (q_p, m_p)$ as

$$\partial_\mu j_p^\mu = \partial_t q_p - \nabla_p m_p = i[H_\partial, q_p] - (m_{p+1} - m_{p-1}) = 0, \tag{26}$$

where $\nabla_p$ is the discrete derivative in real space. As the charge of the symmetry is given by $\mathcal{W}$, $q_i$ should be given as $q_i = (w_i + w_{i+1})/3$. The sum is taken to ensure space reflection symmetry of $q_i$, which is broken for $w_i$. Using the original form of edge Hamiltonian, one can check that $i[H_\partial, w_p/3] = m_p - m_{p-1}$ if

$$m_p = i\bar{X}_p \left( \frac{1}{3}\delta_{(n_{p-1}-n_{p+1})(n_{p-1}+n_p+n_{p+1}+1)} - \delta_{n_{p-1}-n_{p+1}}\delta_{n_{p-1}+n_p+n_{p+1}+1} \right) + \text{H.c.} \tag{27}$$

and Eq.26 follows immediately. In the process of derivation, we have used identities $\delta_{ab} = \delta_a + \delta_b - \delta_a\delta_b$, $\delta_a\delta_b = \delta_a\delta_{ka+b}$ and $\delta_n + \delta_{n+1} + \delta_{n-1} = 1$ for any $a, b, k, n \in \mathbb{Z}$.

## 3.2 Boundary 't Hooft anomaly for the SPT Hamiltonian

In the nontrivial SPT phase, the Hamiltonian we found at the boundary for the non-trivial edge mode possesses the anomalous $\mathbb{Z}_3$ symmetry itself called 't Hooft anomaly [62]. Generally speaking, the 't Hooft anomaly is the obstruction to introducing a gauge symmetry with a given discrete group into the system, which results in irreducible topological terms. In our case, it is the initial $\mathbb{Z}_3$ symmetry. Let us consider the uniform symmetry operator $S$ to see the anomaly. The transformation of Eq.5 affects the symmetry group $\{\mathbb{I}, S, S^\dagger\}$ as well: $\mathbb{I} \to \mathbb{I}$, $S \to V^\dagger S$, $S^\dagger \to S^\dagger V$. Note that in this section, we only consider the boundary Hamiltonian, so the subscript "$\partial$" is dropped everywhere.

The presence of an anomaly indicates that we should pass to the projective representation of the group based on the modification of the associativity condition of group multiplication. The physical picture for the non-trivial SPT phases looks as follows [64]: the ground state of the system consists of different regions separated by domain walls, representing defect lines of the discrete symmetry realizations. Moreover, the states on the defect lines themselves are the realization of SPT phases in one lower dimension.

As a result of our symmetrization procedure, the factor $S = \prod_p X_p$ commutes trivially with the Hamiltonian. One can check that $V$ also commutes with the Hamiltonian and $S$. This follows from the fact that the exponent of $V$ (Eq.12) tracks the passes between sites with states $n = 1$ and $n = 2$ over the boundary. Thus $V = \varepsilon^{\mathcal{W}}$ for the closed interval, where $\mathcal{W}$ is the winding number symmetry (Eq.24). In this sense, $S$ and $V$ form a $\mathbb{Z}_3 \times \mathbb{Z}_3$ symmetry group. Summing up, the elements of the transformed symmetry group can be given by

$$\mathcal{S}(g) = V^{-g}S^g = \left( \prod_p \varepsilon^{-g n_p n_{p+1}(n_{p+1}-n_p)} \right) \cdot \left( \prod_p X_p^g \right), \tag{28}$$

with $g \in \{0, 1, 2\}$. Now, let us identify the anomaly of this symmetry. Once the symmetry is considered on a finite section of the edge, the anomaly manifests itself as broken associativity at the endpoints of this section. The implied locality of $\mathcal{S}$ allows us to consider a single endpoint at a time [65,66]. First, we restrict the symmetry operators to a half-infinite interval $p \in (0, 1, \dots)$

$$\mathcal{S}_r(g) = \left( \prod_{p=0}^\infty \varepsilon^{-g n_p n_{p+1}(n_{p+1}-n_p)} \right) \cdot \left( \prod_{p=0}^\infty X_p^g \right). \tag{29}$$

These operators satisfy

$$\mathcal{S}_r(g)\mathcal{S}_r(h) = A(g,h)\mathcal{S}_r(gh), \qquad A(g,h) = \varepsilon^{hgn_0^2 + hg^2 n_0}. \tag{30}$$

The anomaly is then given by the extra phase factor, $\omega_3$, in the associativity condition

$$\mathcal{S}_r(g)(\mathcal{S}_r(h)\mathcal{S}_r(k)) = \omega_3(g,h,k)(\mathcal{S}_r(g)\mathcal{S}_r(h))\mathcal{S}_r(k), \tag{31}$$

which indicates that $\mathcal{S}_r$ is a projective representation of the $\mathbb{Z}_3$ symmetry group. The 3-cocycle $\omega_3$ can be calculated explicitly as

$$\omega_3(g,h,k) = \frac{\mathcal{S}_r(g)A(h,k)\mathcal{S}_r^{-1}(g)A(g,hk)}{A(g,h)A(gh,k)} = \varepsilon^{ghk(g+h)}. \tag{32}$$

This is precisely the non-trivial element of cohomology group $H^3(\mathbb{Z}_3, U(1))$ that we started with.

# 4 Alternate forms of the Hamiltonian

The boundary Hamiltonian can be further simplified upon introducing the "wall operators," $\hat{n}_p$, and the corresponding $\hat{X}_p$ to satisfy

$$\begin{aligned} \hat{n}_p &= n_{p+1} - n_p \quad \mathrm{mod}\ 3, \\ \bar{X}_p &= \hat{X}_{p-1}\hat{X}_p^\dagger. \end{aligned} \tag{33}$$

These operators straightforwardly obey the initial algebra of operators $n_p$ and $X_p$. In terms of operators in Eq.33, $H_\partial$ can be written as

$$H_\partial = -\sum_{p\in\partial} \hat{X}_{p-1}\hat{X}_p^\dagger \delta_{\hat{n}_{p-1}+\hat{n}_p} + \delta_{\hat{n}_{p-1}}\hat{X}_{p-1}\hat{X}_p^\dagger \delta_{\hat{n}_p} + \delta_{\hat{n}_p}\hat{X}_{p-1}\hat{X}_p^\dagger \delta_{\hat{n}_{p-1}} + \text{H.c.}, \tag{34}$$

which is the easiest to verify by directly observing the action of different terms of Eq.16 on possible configurations $\{n_p\}$. This can be further transformed into the following form

$$H_\partial = -\sum_{p\in\partial}(\hat{X}_{p-1}\hat{X}_p^\dagger + \text{H.c.})\delta_{\hat{n}_{p-1}+\hat{n}_p} - (\delta_{\hat{n}_p} - \delta_{\hat{n}_{p-1}})(\hat{X}_{p-1}\hat{X}_p^\dagger + \text{H.c.})(\delta_{\hat{n}_p} - \delta_{\hat{n}_{p-1}}). \tag{35}$$

Here we used the fact that $\delta_{n_p}X_p\delta_{n_p} = 0$. The expression can be written in a more compact form by introducing a notation for a discrete transfer operation: $(1+\Delta)A_p = A_{p+1}$, for any operator $A_p$. Note that $\Delta$ is the discrete derivative to the right, and $(1+\Delta)^{-1} \neq (1-\Delta)$, as it would be in the infinitesimal continuous case. Then, the boundary Hamiltonian acquires the following form:

$$H_\partial = -\sum_{p\in\partial}(2 - \hat{X}_p\overleftrightarrow{\Delta}\hat{X}_p^\dagger)\delta_{(\Delta-1)\hat{n}_p} + \Delta\delta_{\hat{n}_p}(2 - \hat{X}_p\overleftrightarrow{\Delta}\hat{X}_p^\dagger)\Delta\delta_{\hat{n}_p}. \tag{36}$$

This can be used in studies of the ground state properties and the excited states of the boundary model.

## 4.1 From $\mathbb{Z}_3^N$ to $\mathbb{Z}^N/3\mathbb{Z}$

The Hilbert space of our boundary model, $\mathcal{H}_\partial$, has a basis labeled with strings of $N$ numbers $n_i \in \{-1, 0, 1\}$. We could expand this model to one set in a bigger Hilbert space, $\tilde{\mathcal{H}}_\partial$, where basis elements are strings of $N$ arbitrary integers, up to a total shift by 3:

$$\mathcal{H}_\partial = L^2(\{|n_1, \ldots n_N\rangle : n \in \mathbb{Z}_3^N\}) \hookrightarrow \tilde{\mathcal{H}}_\partial = \frac{L^2(\text{span}\{|n_1, \ldots, n_N\rangle : n \in \mathbb{Z}\})}{S^3}. \tag{37}$$

Now, one can define operators $X_p$, $X_p^\dagger$ on $\tilde{\mathcal{H}}_\partial$ similarly to what was done for $\mathcal{H}_\partial$. They act on the states at $p$ by increasing or decreasing $n_p$ by 1. We already used this redefinition of $X_p$ in Eq.37, as a part of $S = \prod_p X_p$. In the expanded Hilbert space $\tilde{\mathcal{H}}_\partial$, operators $X_p$ and $n_p$ form another representation of the ladder algebra because the relation $X_p^3 = 1$ is lost.

There is an immersion $\mathcal{I} : \mathcal{H}_\partial \hookrightarrow \tilde{\mathcal{H}}_\partial$, which maps the basis elements of $\mathcal{H}_\partial$ to the basis elements of $\tilde{\mathcal{H}}_\partial$ in the following manner. Let $\mathcal{I}(|n_1, \ldots, n_N\rangle) = |\tilde{n}_1, \ldots, \tilde{n}_N\rangle$, where

$$\begin{aligned} \tilde{n}_1 &= n_1, \\ \tilde{n}_k &= \tilde{n}_{k-1} + w_k, \qquad k \geq 2. \end{aligned} \tag{38}$$

Reversing the procedure, we note that there is a surjective map $\mathcal{S} : \tilde{\mathcal{H}}_\partial \twoheadrightarrow \mathcal{H}_\partial$ that takes $|\tilde{n}_i\rangle$ to $|\tilde{n}_i \bmod 3\rangle$. An operator $\mathcal{P} = \mathcal{I}\mathcal{S}$ is a projection operator on $\text{Im}\,\mathcal{I}$ and $\mathcal{S}\mathcal{I} = \mathbf{1}$.

To this end, one can define a Hamiltonian acting on the larger Hilbert space $\tilde{\mathcal{H}}_\partial$ as follows:

$$\tilde{H}_\partial = -\sum_{p \in \partial} \left[ X_p^2 \delta_{n_p - n_{p-1}, -1} \delta_{n_{p+1} - n_p, +1} + X_p (1 - \delta_{n_p - n_{p-1}, +1})(1 - \delta_{n_{p+1} - n_p, -1}) \right] + \text{H.c.} \tag{39}$$

It has a few special properties. First, the subspace $\text{Im}\,\mathcal{P} \approx \mathcal{H}_\partial$ is invariant under $\tilde{H}_\partial$

$$[\mathcal{P}, \tilde{H}_\partial] = 0, \tag{40}$$

and the restriction of $\tilde{H}_\partial$ to $\mathcal{H}_\partial$ is $H_\partial$:

$$H_\partial = \mathcal{S}\tilde{H}_\partial\mathcal{I}. \tag{41}$$

Second, it is defined in terms of ladder operators that act on the space of states labeled by $\mathbb{Z}$-numbers instead of $\mathbb{Z}_3$.

As the next step, let us closely look at what the basis states in $\text{Im}\,\mathcal{P}$ look like. Those are labeled by sets of numbers $n_1, \ldots, n_N$ that satisfy

$$|n_p - n_{p-1}| \leq 1, \qquad 2 \leq p \leq N, \tag{42}$$

up to a total shift of 3. The fact that $\text{Im}\,\mathcal{P}$ is conserved by the Hamiltonian means that if we start with a state that satisfies the condition Eq.42, we will obtain a linear combination of states that also satisfy this condition. For those states, the winding number is just

$$\mathcal{W} = \frac{n_N - n_1 + \delta_{n_1 - n_N - 1} - \delta_{n_1 - n_N + 1}}{3} = \left[ \frac{n_N - n_1 + 1}{3} \right], \tag{43}$$

where $[\ ]$ denotes the integer part. We can make this statement even stronger. The Hamiltonian acting on a basis state $|n_i\rangle$ that satisfies the condition Eq.42 produces a linear combination of all states that differ from $|n_i\rangle$ in exactly one spin, satisfy Eq.42 and have the same winding number. This proves that there are no diagonal symmetry operators other than $\mathcal{W}$, as any two states $|\Psi_1\rangle, |\Psi_2\rangle \in \mathcal{H}_\partial \approx \text{Im}\,\mathcal{P}$ with the same winding number produce a nonzero matrix element of the evolution operator:

$$\langle \Psi_2 | \exp(-iH_\partial t) | \Psi_1 \rangle \neq 0. \tag{44}$$

This shows that the only symmetry of our Hamiltonian which is diagonal in basis of $n_i$ operators is $\mathcal{W}$.

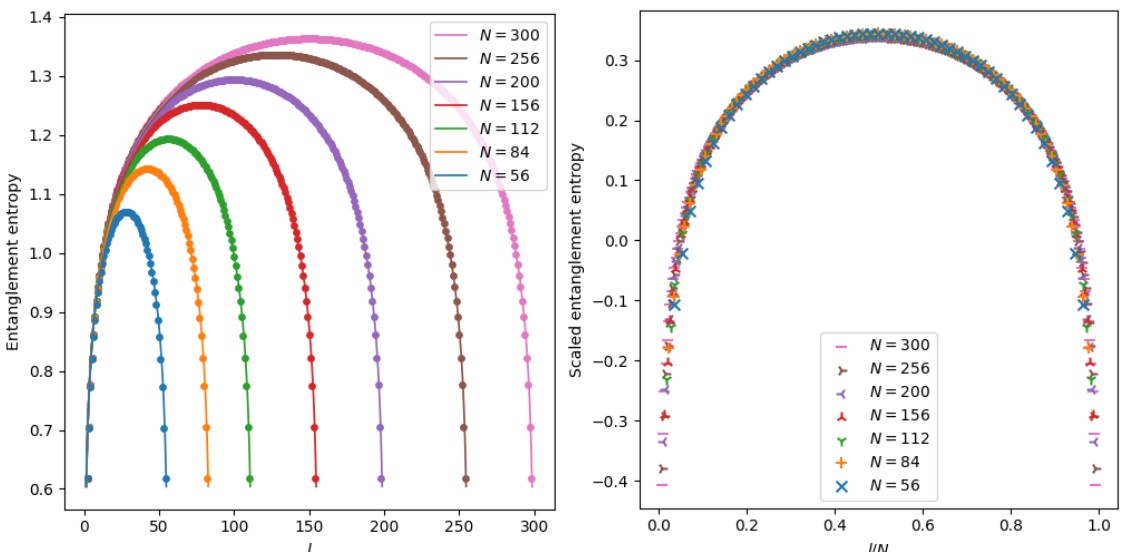

Figure 2: Points represent numerical values of the unscaled (left) and scaled (Eq.46) (right) entanglement entropy for finite open chains with lengths $N = 56, 84, 112, 156, 200, 256, 300$. Curves are defined by Eq.45 with $c = 1.06$ and $a = 0.557$.

## 5 Conformal properties of the edge model

Similarly to our previous study, Ref. [61], we detect conformal properties of the edge modes by analyzing the finite-size effects of the gap and entanglement entropy of the boundary model (Eq.15). For numerical calculations, we use the transformed form (using Eqs. A.1-A.3) of the Hamiltonian.

To find the central charge of the massless edge excitations, we should analyze the entanglement entropy of the ground state. We perform calculations of the entanglement entropy for open and closed chains. The results of the numerical calculations for open chains of lengths $N = 56, 84, 112, 156, 200, 256, 300$ are presented in Fig.2. As shown in Ref. [67], the entanglement entropy in the conformal field theory with a central charge $c$ on the open strip of length $N$ can be determined by the following formula:

$$S_N(l) = a + \frac{c}{6} \log\left(\frac{N}{\pi} \sin\left[\frac{\pi l}{N}\right]\right).$$  (45)

Here, $a$ is a constant while $l$ is the length of entangled spins. For a closed chain, the proportionality coefficient should be set to $c/3$ instead of $c/6$. This equation characterizes how the entanglement entropy of the spin chain model at criticality changes with the finite system size. The numerical values for the entanglement entropy of the open chains at various values of $N$ are shown in Fig.2. It also demonstrates the fitting of the analytic formula Eq.45 with parameters $c = 1.064$ and $a = 0.557$, as well as the collection of all numerical points of open chains in a single scaled function.

$$S_{scaled}(x) = S_N(l) - \frac{c}{6} \log[N] = a + \frac{c}{6} \log\left(\frac{1}{\pi} \sin[\pi x]\right), \qquad x = \frac{l}{N}, \quad l = 2, \dots, N-1.$$  (46)

Similar results for closed chains are presented in Fig.3. Here, we have analyzed smaller chain lengths because, in this case, the calculations take longer. Fitting the results to Eq.46 but with coefficients $\frac{c}{3}$ in front of log we have obtained $c = 1.01$ and $a = 0.991$. Both results unambiguously show that our edge model in the continuum limit is described by a CFT with

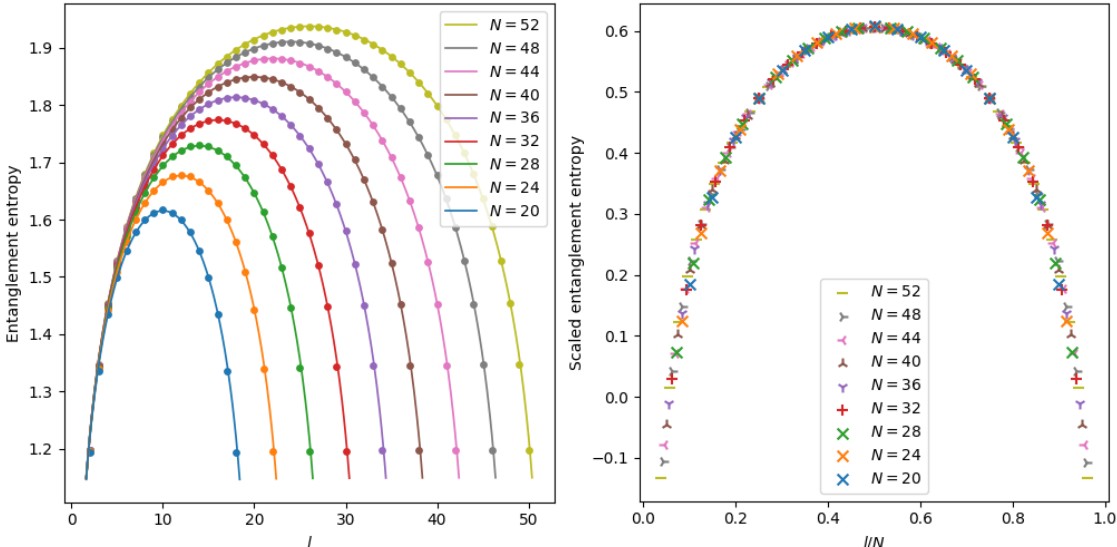

Figure 3: Points represent numerical values of the unscaled (left) and scaled (right) entanglement entropy for finite closed chains with lengths $N = 20, 24, ..., 52$. Curves are defined by Eq.45 but with coefficient $\frac{c}{3}$ instead of $\frac{c}{6}$, with $c = 1.01$ and $a = 0.991$.

a central charge $c = 1$. Thus, it may be tempting to conclude that the edge model for $\mathbb{Z}_3$ paramagnets coincides with the edge states of the Levin-Gu model with $\mathbb{Z}_2$ symmetry, which is defined by the XX model of free fermions [12]. However, this is not the case, as shown below.

We then use the DMRG approach to compute the excitation gap in our model for various system sizes, $N$. According to [68,69], the finite-size behavior of the lower energy states reads

$$E_0(N) = kN + b - \frac{\pi v c}{6N} + \mathcal{O}(N^{-2}), \qquad E_i(N) = E_0(N) + \frac{2\pi v x_i}{N} + \mathcal{O}(N^{-2}), \qquad (47)$$

where $k, b, v$ are some constants, $c$ is the central charge, and $x_i = h_i + \bar{h}_i$ is the scaling operator dimension (sum of holomorphic and anti-holomorphic scaling dimensions of primary fields $h_i$ and $\bar{h}_i$) of the corresponding CFT. From the calculated values of the energies, we obtain the numerical value for the leading scaling dimension $x_1 \approx 0.95$, which corresponds to $x_1 = 1$ within the available precision. The leading scaling dimension constitutes the correlation length. The numerical data for the energies are presented in Fig.4. Calculations have been made for an open system of sizes $N = 16 \sim 244$. Using exact diagonalization for smaller system sizes (up to N=14), we have also calculated the spectrum of the first excited states and found that at the gap point, the momentum is zero. It supports our expectation that conformal dimensions of the primary fields $h = \bar{h} = 1/2$, but further confirmation of this statement is necessary at larger system sizes.

Another piece of information about the low-energy effective CFT can be obtained from the study of Kac-Moody currents, which can appear in the model. The winding number symmetry $\mathcal{W}$ forms a $U(1)$ current $j_p^\mu = (q_p, m_p)$ and it is necessary to check whether it can be holomorphically factorized into chiral $U(1)$ currents, $j_p^\pm = q_p \pm m_p$, forming a Kac-Moody algebra. For the latter to form, the currents must satisfy commutativity relations,

$$[j_p^+, j_{p'}^-] = 0, \qquad (48)$$

in the thermodynamic limit. This is necessary for the currents to be independent. Using the exact diagonalization approach, we numerically calculate matrix elements of the commutators mentioned above between low-energy states for relatively short system sizes. Our results for

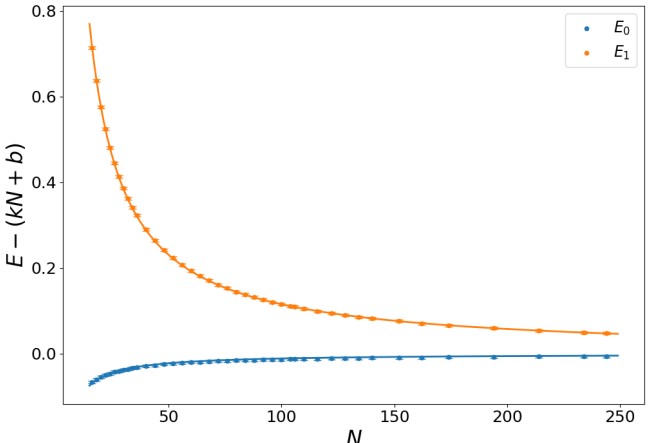

Figure 4: The edge mode's ground state energies $E_0$ and first exited state energies $E_1$ (excluding the linear term $kN + b$ (Eq.47)) versus the boundary length. Calculations are done on open chains by simulation based on density matrix renormalization group (DMRG). The error bars demonstrate the precision of calculations and are of the order $3 \cdot 10^{-3}$. The lines are the $\propto N^{-1}$ fitting curves.

systems of sizes $N = 4, 6, ..., 14$ show that commutators with $p' = p$ are numerically 0. In the $p' = p + 1$ case, they decrease as $\sim N^{-1.35}$, and are exactly 0 otherwise. The calculations are presented in Fig.5.

The next step is to check the holomorphicity condition:

$$\partial_- j^+ = \partial_+ j^- = 0, \tag{49}$$

with $\partial_\pm = \partial_t \pm \nabla_p$. After excluding the analytically known relation $\partial_- j^+ + \partial_+ j^- = 2\partial_\mu j^\mu = 0$ we are left with equation $\partial_- j^+ - \partial_+ j^- = 0$. The latter can be written as

$$i[H, m_p] - (q_{p+1} - q_{p-1}) = 0, \tag{50}$$

in our discrete case. We use the same numerical approach to test the validity of the conjecture in the *thermodynamic* limit. The dependence of matrix elements for the both commutator and difference terms on the system size $N$ for $N \in \{3, 4, ..., 15\}$ is presented in Fig.5. It has some instabilities, but it is clear that their upper limit exponentially decreases as $e^{-N/6}$, implying the same decay for the overall expression.

Finally, after verifying the existence of Kac-Moody currents, we have to check the corresponding anomaly, given by the momentum space current commutators $[j_n^\pm, j_{-n}^\pm] = nk$ with $j_n^\pm = \frac{1}{2\pi} \sum_{p=1,N} e^{\frac{\pi i n p}{N}} j_p^\pm$. $k$ here denotes the level of the Kac-Moody algebra and defines the anomaly. The commutator can be reduced to a simpler form using the fact that $[j_p^\pm, j_{p'}^\pm] \neq 0$ if and only if $p' = p \pm 1$. Thus

$$[j_n^+, j_{-n}^+] = \frac{i}{2\pi^2} \sin\left[\frac{2\pi n}{N}\right] \sum_p [j_p^+, j_{p+1}^+] \xrightarrow[n \ll N]{} n \frac{i \sum_p [j_p^+, j_{p+1}^+]}{\pi N}. \tag{51}$$

Resorting again to numerics, we are able to determine the Kac-Moody algebra level. Computations similar to the one above, produce $k \simeq 0.95$, which reproduces $k = 1$ within the precision of $O(10)^{-2}$ as shown in Fig.6.

Based on the values of $c$, $x_1$, and $k$, it can be argued that the corresponding edge theory can be given by the $SU_1(3)/SU_1(2)$ coset CFT model of current algebras at level k=1. $SU_1(3)$ contains two $U_1(1)$ subgroups at level k=1, one of which will be gauged out as a subgroup

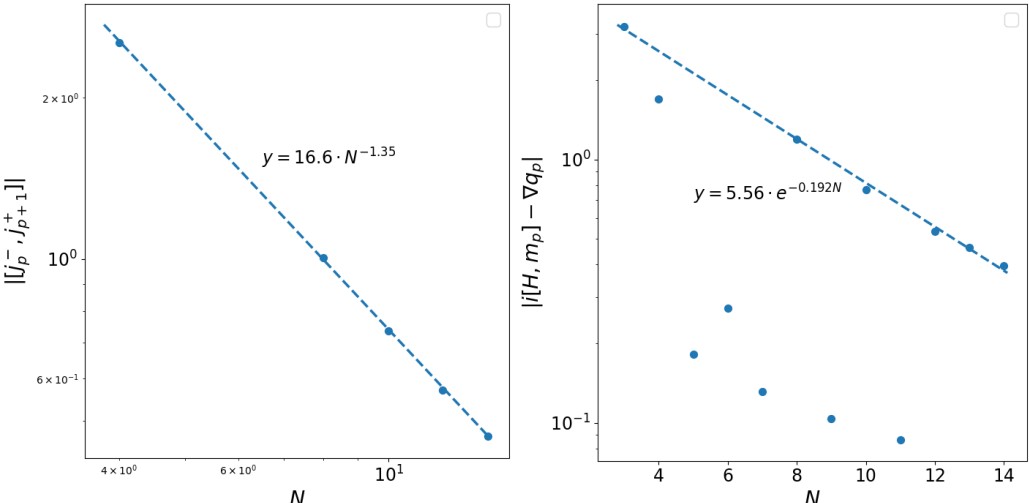

Figure 5: The finite-size behavior of expressions indicating the existence of anomalous $U(1)$ symmetry of the boundary Hamiltonian. The left panel shows a power law decrease of the commutator, $[j_p^+, j_{p+1}^-]$. The right panel shows an exponential decrease of $i[H, m_p] - \nabla q_p$.

of the denominator group $SU_1(2)$, while the next will remain. It is clear that this $U(1)$ is the maximal remaining subgroup in the factor and is connected to the anomalous winding number symmetry observed above as $U(1)$ Kac-Moody at level k=1. Moreover, one can expect that 't Hooft anomalous $\mathbb{Z}_3$ presented in our boundary model (Eq.16) can be a subgroup of this remaining $U_1(1)$ Kac-Moody, since a central extension in a current algebra can be regarded as a projective representation of $U(1)$ symmetry. This means that a phase factor will appear in the partition function of our boundary model after $U(1)$ symmetry transformation, as 't Hooft $\mathbb{Z}_3$ anomaly should appear. Of course, precise observation of this phase factor is necessary, which is a task for subsequent studies. The 't Hooft anomaly in the low-energy model can also be detected in other ways, as presented in Refs. [65, 70]. As for the boundary model's global $S_3$ symmetry (Eq.19), it also exists in the $SU(3)/SU(2)$ coset CFT: the $SU(3)$ group has three independent (despite overlapping) $SU(2)$ subgroups, which allows different ways to make the $SU(2)$ gauging, yet leading to the same CFT. Thus we have a global $\mathbb{Z}_3$ symmetry of choice between the three $SU(2)$ subgroups. Together with the $\mathbb{Z}_2$ conjugation symmetry, it forms an $S_3$ group.

# 6 Concluding remarks

We have studied the three-state paramagnetic Potts model with $\mathbb{Z}_3$ symmetry on a 2D triangular lattice. As expected, the different SPT phases are classified according to the cohomology group $H^3(\mathbb{Z}_3, U(1)) = \mathbb{Z}_3$. We have constructed the unitary operator responsible for this phase and found the Hamiltonian of edge states. The numerical study of the finite-size effects of the edge Hamiltonian showed that it is gapless and that the conformal dimension of the scaling operator, which defines the correlation length, is $x_1 = 1$. Calculations of the entanglement entropy of the low-lying excitation show that it has a central charge $c = 1$. We also find a hidden $U(1)$ symmetry of the Hamiltonian corresponding to the winding number, which, as it appeared, leads to an anomalous $U(1)$ Kac-Moody current algebra with level k=1. This analysis emphasizes that the effective theory of low-energy excitations of our edge Hamiltonian is the coset CFT $SU_k(3)/SU_k(2)$ with k=1. This coset contains the $U(1)$ anomalous subalgebra.

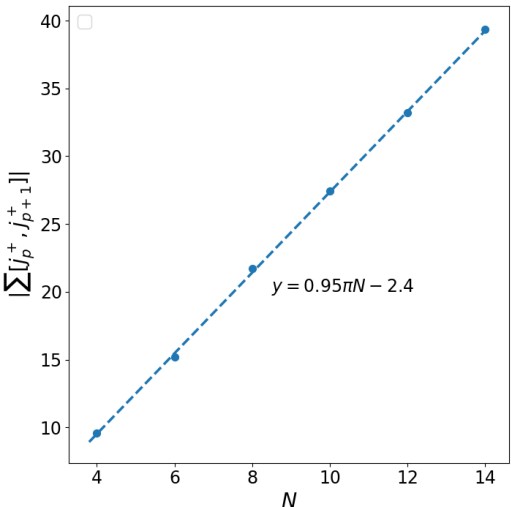

Figure 6: $|\sum_p [j_p^+, j_{p+1}^+]|$ versus system size $N$. The slope of the line is $0.95\pi$, indicating that current anomaly $k \approx 1$.

The incorporation of SPT degrees of freedom into the three-state Potts paramagnet opens new avenues for understanding phase transitions, anyonic excitations, and the interplay between topology and quantum information. Experimental realizations of the SPT Potts paramagnet, such as in the cold atom systems or magnetic materials, are still an open problem. The presented work opens the possibility of developing experimental platforms for probing and manipulating the topological aspects of $\mathbb{Z}_3$ SPT paramagnets.

## Acknowledgments

The authors are grateful to A. Belavin, A. Litvinov, Sh. Khachatryan and H. Babujian for helpful discussions.

**Funding information** The research was supported by startup funds from the University of Massachusetts, Amherst (TAS) and Armenian SCS grants Nos. 21AG-1C024 (MM, AS), 21AG-1C047 (TH), 24RL-1C024 (HT), and 24FP-1F039(TH, HT, AS).

## A Identities

One can check that the following identities hold for $n_1, n_2, n_3 \in \{-1, 0, 1\}$ and $\varepsilon = e^{2\pi i/3}$

$$\varepsilon^{n_1 n_2 (n_2 - n_1)} = \frac{1}{3}\left(\varepsilon^{n_1 - n_2} - \varepsilon^{n_1 + n_2} - \varepsilon^{-n_1 - n_2} + \varepsilon^{-n_1} + \varepsilon^{n_2} + 2\right), \tag{A.1}$$

$$\varepsilon^{n_1 n_2} = \frac{1}{3}\left(1 + \varepsilon^{n_1} + \varepsilon^{-n_1} + \varepsilon^{n_2} + \varepsilon^{-n_2} + \varepsilon^{n_1 + n_2 - 1} + \varepsilon^{-n_1 - n_2 - 1} + \varepsilon^{1 + n_1 - n_2} + \varepsilon^{1 - n_1 + n_2}\right), \tag{A.2}$$

$$1 + \varepsilon^{(n_1 - n_3)(1 + n_1 + n_2 + n_3)} + \varepsilon^{-(n_1 - n_3)(1 + n_1 + n_2 + n_3)}$$

$$= \frac{1}{3}\left[\frac{5}{2} - \varepsilon^{1 - n_1 + n_2} - \varepsilon^{1 - n_3 + n_2} + 2\varepsilon^{n_1 - n_3} + 2\varepsilon^{1 + n_1 + n_2 + n_3}\right] + \text{c.c.} \tag{A.3}$$

These identities can be used to reduce two-site interactions of given forms to the sum of single-site operator products. It proves useful for computational purposes.

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
