# Peer review of "Two-dimensional topological paramagnets protected by Z3 symmetry: Properties of the boundary Hamiltonian"

_SciPost Physics, doi:SciPost Phys. 18, 068 (2025)_

## Round 2 · Referee Report · Anonymous (Referee 2) · 2025-1-4

Report

The authors have very nicely addressed various suggestions/comments in the referee report. This manuscript meets the expectations and criteria for publication in SciPost. However, before that I still have two comments/questions regarding the updated manuscript, and I greatly appreciate the patience of the authors and editors.

  1. At the end of Section 2, the authors explained that the two non-trivial $\mathbb{Z}_3$ SPT phases have the same edge Hamiltonian. On the other hand, however, we do expect that these two edge theories have different $\mathbb{Z}_3$ anomaly values. Perhaps I am missing something, but I am confused about how to reconcile these two statements. It would be nice if the authors could comment a little bit on this.

  2. Regarding the leading total scaling dimension corresponding to the first excited state: if this scaling dimension is smaller than $1$, then we know for sure that the corresponding CFT operator is a primary operator, which under the assumption that it has spin 0 would have both holomorphic and anti-holomorphic conformal weight equal to $1/2$ of the total scaling dimension. However, if this scaling dimension is equal to $1$, then we can't rule out the possibility that the corresponding operator is the first descendant of the identity operator, unless one also analyzes the momentum data capturing the difference between the holomorphic and anti-holomorphic conformal weights, using the relation (for periodic boundary condition) \begin{equation} P=\frac{2\pi}{N}(h-\bar{h})+O(\frac{1}{N^2})~. \end{equation}

I understand that the authors have been using open boundary conditions for computational purposes, and I am not demanding the authors to further extract $h-\bar{h}$. However I do want to point out the potential loophole regarding this point.

Recommendation

Ask for minor revision

---

## Round 2 · Referee Report · Anonymous (Referee 1) · 2025-1-25

Report

The authors addressed comments of referees and substantially improved the manuscript. I now recommend publication in Scipost Physics.

Recommendation

Publish (easily meets expectations and criteria for this Journal; among top 50%)

---

## Round 2 · Author Response

We sincerely appreciate the time and effort the referees dedicated to reviewing our work and for their constructive feedback.

We have carefully considered all the comments and suggestions provided by the referees and have addressed them thoroughly in the revised version of the manuscript. Additionally, we have prepared a detailed response to the referees’ questions and comments, which are included below for your convenience.

Our revisions have significantly strengthened the manuscript, and we hope it now meets the high standards of SciPost Physics.

Best regards,

The authors.

================= Responses to referee comments =================

We thank the referees for their efforts and appreciate the comments, many of which have made the revised paper more complete and more straightforward to comprehend for readers. Below, we address all the points raised by the referees.

Our response to Referee 1:

  1. We would disagree that the current model is closely related to the one studied in [61]. Of course, the model resulting from the Z_3 symmetry is also induced in the case of the (Z_3)^3 symmetry. However, it was neither formulated nor studied in [61]. The approach to the investigation of the model is indeed similar to [61], there are new obstacles here that have to be overcome. One such example is the anti-symmetrization procedure of the relevant cohomology group element. The paper has a more generalizable model formulation approach than [61]. These points are now more clearly stated in the paper.

  2. A figure is now added for easier comprehension, as suggested.

3.1. The question about different edge theories is now addressed in the paper (the end of section 2). There are three phases, but all the non-trivial boundary models are the same.

2, 3.2, 5. For clarity, we have reformulated some of the ideas discussed in the manuscript (e.g., the one mentioned in the abstract, for the phrase after eq.25, etc.) and made some text rearrangements (section 5) for a more coherent narrative structure. Additionally, we included extra comments regarding the formula derivations.

  1. The question of embedding the S_3 symmetry on the boundary CFT is now addressed at the end of section 5.

Our response to Referee 2

  1. A figure is now added as a part of the definition for "color" and generally for easier comprehension. The $p_p$ typo is now fixed (it should have been $n_p$).

  2. The two non-trivial edge theories result in the same edge Hamiltonian as the first one, as they are induced by transformations U and its conjugate U^+. The third boundary phase is the trivial one. A more detailed discussion is added at the end of section 2.

  3. The formulation in the abstract about tricriticality is now made more transparent, and a discussion of it is added after eq.19.

4.1. The mismatch of the formula and the figure boils down to the fact that the Hamiltonian is defined up to a positive coefficient. Thus, there is an energy scale freedom. The figure is now updated and contains more information. The mismatch is fixed. The text around eq.45 (now eq.47) has also been refactored.

4.2. The choice of open boundary conditions is purely computational, as it is significantly more challenging for the computer to calculate closed chains.

4.3. The computed value x=1 is the (total) scaling dimension of the leading scaling operator, which (assuming that holomorphic and anti-holomorphic components have the same dimension) brings the dimension of each component to 1/2. The complete study of the primary field set in the SU(3)/SU(2) coset CFT and their conformal dimensions is a matter of future work.

  1. Computed values of central charge, CFT primary field dimension, anomalous current algebra level, and symmetry analysis support the proposed CFT. We consider further justification outside of the scope of this particular paper, and it is a topic that we plan on studying further.

  2. The fixes have been done.

Our response to Referee 3

Section 4. The symmetries indeed help to understand the boundary theory. First, some symmetries (namely the winding number) are a feature of the boundary model and don't exist otherwise. The discrete symmetry indicates gapless excitations. Then, during the identification of the continuum limit theory, they are also helpful, as the symmetries should also transfer to the continuum limit. The system ’t Hooft anomaly is also related to those symmetries.

Numerical calculations are also necessary. They provide information about the system, including its central charge, corresponding CFT lightest primary field scaling dimension, and anomalous current algebra level. The last one is directly connected to the symmetries, as the expressions of those currents are directly derived from symmetry expressions. Moreover, the numerical calculations would be much more efficient if Hamiltonian's symmetries were explicitly known.

Section 5. It is impossible to determine the central charge from the energy gap analysis as it always appears next to the velocity coefficient v. This part of the paper was revised, and everything is written explicitly now.

Although starting with the Potts model, the derived edge theory differs from the conventional Potts model in 1D. As the paper states, the theory is anomalous and incompatible with parafermions.

Overall, we have done some idea reformulations for clarity and made text rearrangements (section 5) in the manuscript for a more coherent narrative structure. Additionally, we included extra comments regarding the formula derivations and the phenomenon of physics in general.

---

## Round 2 · List of Changes

• The sentence concerning the tricriticality is made more clear.

  • An explanation of the relation between the Z_3 and Z_3^3 models is added in the last paragraph of the Introduction.

  • A figure displaying the lattice (Fig.1) is added to support the definitions and the boundary model derivation.

  • Eq.4 and the text around it are modified to make the derivation of the unitary transformation easier to understand.

  • The second-to-last paragraph in Section 2 is added to explain how considering a single Hamiltonian covers all the existing boundary phases.

  • The last paragraph (sentence) in Section 2 is added to emphasize the novelty of the current derivation approach.

  • A paragraph is added after Eq.19 to explain the mechanism of the tricritical symmetry in detail.

  • The notion of the Winding number symmetry being a "distinctive feature" is clarified before Eq.26.

  • The order of discussing the entanglement entropy and the first excitation gap is switched in section 5.

  • The analysis of the first excitation gap (Eq.47 and the surrounding text) is now more detailed and comprehensible.

  • The figure for the first excitation gap (Fig.4) is changed to fix the energy-scale inconsistency and to match the new narrative structure.

  • The last paragraph of Section 5 is modified to include the discussion of embedding the S_3 symmetry of the initial boundary model in the proposed CFT.

  • Other minor (1-2 word) changes are done throughout the manuscript for grammar and aesthetic reasons.

---

## Round 3 · Author Response

We thank the Referees for their positive reports.

Below, we address Report #1 by Referee 3:

  1. The Hamiltonian Eq. (5) describes Nontrivial SPT states created by the unitary operators U and U^+, defined in Eq.(4). Combined with the identity operator, U and U^+ form the cohomology group Z_3. The trivial state, |0>, is the ground state of the Hamiltonian (5), which has a matrix product form. The two SPT states represent are connected with each other by |0'> = U |0>, |0''> = U^+ |0> = U |0'>.

Each SPT state has edge modes that restore the Z_3 symmetry in the presence of a boundary. The corresponding boundary Hamiltonian supporting these modes is defined in Eq.(7), where V and V' are defined by values of cohomological nontrivial U (or U2) on the boundary or trivial identity. Replacing U with its topologically different U^+, we replace V and V' with their Hermitian conjugates, which doesn't change the boundary Hamiltonian. The situation is different for the energy eigenstates. The edge states are connected with each other by the operator S acting on the boundary. They are gapless but differ from each other (for example, one can think of two types of spinons or magnons in s=1/2 systems), representing different edge states of the two topologically nontrivial SPT phases.

  1. Using exact diagonalization for smaller system sizes (up to N=14), we have confirmed that the first excited states at the gap point have momentum 0. Hence, it supports the claim h=\bar{h}. We have included this information in the paper. However, future investigation of a larger system size is necessary.

The revised manuscript now addresses both questions. We hope these explanations are satisfactory.

Sincerely, The authors.

---

## Editorial Decision

published